# Admission to psychiatric hospital for mental illnesses 2 years prechildbirth and postchildbirth in Scotland: a health informatics approach to assessing mother and child outcomes

Julie Langan Martin,[1] Gary McLean,[1] Daniel Martin,[1] Roch Cantwell,[2] Daniel J Smith[1]

[1]Institute of Health and Wellbeing, College of Medical, Veterinary and Life Sciences, University of Glasgow, Mental Health and Wellbeing, Gartnavel Royal Hospital, Glasgow, UK
[2]Perinatal Mental Health Service and West of Scotland Mother and Baby Unit, Leverndale Hospital, Glasgow, UK

**Correspondence to**
Dr Julie Langan Martin; julie.langan@glasgow.ac.uk

## ABSTRACT

**Objective** To identify factors associated with: admission to a specialist mother and baby unit (MBU) and the impact of perinatal mental illness on early childhood development using a data linkage approach in the 2 years prechildbirth and postchildbirth.

**Methods** Scottish maternity records (SMR02) were linked to psychiatric hospital admissions (SMR04). 3290 pregnancy-related psychiatric admissions for 1730 women were assessed. To investigate factors associated with MBU admission, the group of mothers admitted to an MBU were compared with those admitted to general psychiatric wards. To assess the impact of perinatal mental illness on early child development, a pragmatic indicator for 'at potential risk of adversity', defined as a child who was recorded as requiring intensive treatment at any time under the health plan indicators (HPI) and/or who had no record of completing three doses of the 5-in-1 vaccine by 12 months was generated. Logistic regression models were used to describe the association between each variable and the risk of admission between those with a history of prior psychiatric admission and those without.

**Results** Women admitted to an MBU were significantly more likely to be admitted with non-affective psychosis (OR=1.97, 95% CI 1.22 to 3.18), affective psychosis (OR=2.44, 95% CI 1.37 to 4.33) and non-psychotic depressive episodes (OR=1.93, 95% CI 1.42 to 2.63). They were less likely to come from deprived areas (OR=0.68 95% CI 0.49 to 0.93). Women with a previous history of psychiatric admission were significantly more likely to be located in the two most deprived quintiles. Almost one-third (29%) of children born to mothers with a pregnancy-related psychiatric admission were assessed as 'at potential risk of adversity.'

**Conclusions** A health informatics approach has potential for improving understanding of social and clinical factors, which contribute to the outcomes of perinatal mental illness, as well as potential adverse developmental outcomes for offspring.

## BACKGROUND

Good maternal mental health is important for normal childhood development.[1] Maternal

| Strengths and limitations of this study |
| --- |
| ► Whole of Scotland childbirth and psychiatric admission data used for analyses, rather than local data only. |
| ► Robust measure of socioeconomic status (Scottish Index of Multiple Deprivation) compared with other studies, which have used educational status. |
| ► These analyses used only psychiatric admission data rather than outpatient psychiatric attendances. |
| ► Our definition of 'at potential risk of adversity' in offspring was a composite and pragmatic measure derived from the limited child health outcome data and there has been some inconsistency in the implementation of this across health boards. |

mental illness can disrupt optimal parenting processes and can adversely affect childhood development, especially emotional development.[2–4] In particular, maternal mental illness may be associated with increased rates of childhood depression[5] and may also affect children's executive[6] and cognitive functioning.[7] Maternal depression both during pregnancy and in the postnatal period is common and is estimated to affect between 10% and 15% of women.[8 9] While the prevalence of postpartum psychosis is relatively low at 1/1000 births,[10] it is known to be associated with severe adverse outcomes, including maternal suicide and infanticide.[11]

The joint admission of mentally ill mothers and their infants was pioneered by Thomas Main in 1948.[12] Since then the UK, Australia and France have acted as leaders in this field.[13 14] Currently in the UK where possible, postpartum women (and those in later pregnancy) with severe mental illnesses such as psychosis or severe depressive disorder are admitted to a specialised mother and baby

unit (MBU). In the West of Scotland, women in the postnatal period who are the primary carer of their baby and thought to require psychiatric hospital admission are discussed with (and where possible assessed by) the local consultant perinatal psychiatrist. Hospital admission is arranged if required. While most admissions are voluntary, MBUs can also accommodate women who require compulsory treatment under the Mental Health Act. The current Scottish Intercollegiate Guidelines Network clinical guidelines for the management of perinatal mood disorder reflect this.[15] It is recognised that women in the early postpartum period (0 to 6 weeks after childbirth) are at particularly elevated risk of requiring psychiatric admission.[16 17] In our previous study, we found that this risk remains elevated for up to 2 years postchildbirth.[16]

MBUs are highly specialised, expensive and limited resources, where expertise in both treatment of psychiatric disorders and childcare are required.[18] Although there are currently 15 in England and 2 in Scotland,[19]access to this specialised service is poorer in many other high-income countries (such as the USA and Canada) and low and middle-income countries despite women appearing to be satisfied with this type of care.[20] Given the importance of adverse childhood experiences in future health,[21 22] perinatal mental illness and the potential impact it has on offspring is a priority area for research and practice.

## AIM OF THE STUDY
The aim of the study was to use a data linkage approach to investigate factors associated with admission to a specialist MBU and the risk to early childhood development in the context of a pregnancy-related psychiatric admission.

## MATERIALS AND METHODS
We used a dataset from the Information Services Division of National Health Service (NHS) Scotland, which included all maternity records (SMR04) between 2005 and 2009, linked to all psychiatric hospital admission records (SMR02) between 2003 and 2011. This dataset has been described elsewhere.[16] However in brief, for each maternity record, any psychiatric admission was reported by week for the 104 weeks prechildbirth and postchildbirth. Admission types were defined by International Classification of Diseases 10th Revision (ICD-10) codes: psychosis-only admissions included 'non-affective psychosis' (F20, F20.3, F20.5, F20.6, F20.8, F20.9, F21X, F22.0, F22.8, F22.9, F23.0, F23.1, F23.2, F23.3, F23.8, F23.9, F24X, F28X, F29X), 'affective psychosis' (F25.0, F25.1, F25.2, F25.9, F30.2, F31.2, F31.5, F32.3, F33.3) and 'postpartum psychosis' (F53.0, F53.1, F53.9); admissions due to a non-psychotic depressive episode included F32.0, F32.00, F32.01, F32.1, F32.10, F32.11, F32.2, F32.8, F32.9, F33.0, F33.00, F33.1, F33.10, F33.11, F33.2, F33.4, F33.8, F33.9. For the category of 'other admissions', we included all other ICD-10 codes recorded.

In total, 3290 pregnancy-related psychiatric admissions for 1730 women were assessed. For deliveries in 2005, psychiatric admissions between 2003 (2 years before) and 2007 (2 years after) were captured. Similarly, for deliveries in 2009, psychiatric admissions from 2007 (2 years before) until 2011 (2 years after) were captured. For each maternity record, any psychiatric admission was reported by week for the 2 year prechildbirth and postchildbirth periods. Early childhood developmental outcomes (assessed by a health visitor at both 10-day and 6–8 week child–health checks) were also available. All statistical analyses were performed in STATA V.13.1.[23]

To investigate factors associated with admission to the MBU, we compared the group of mothers admitted to one of two Scottish MBU units with those admitted to a general psychiatric ward up to 2 years postchildbirth on a range of sociodemographic characteristics (age, social deprivation and previous pregnancy). We included the geographical location (health board area) of the mother. We also compared differences in average length of stay. There are two MBUs in Scotland, the West of Scotland Mother and Baby Unit (Leverndale Hospital, NHS Greater Glasgow and Clyde) and Mental Health Mother and Baby Unit, St John's Hospital (West Lothian) where mothers are admitted with their baby in the postpartum period. For women admitted to a general psychiatric hospital, this is generally without their baby.

To explore possible risk factors associated with previous psychiatric admission in the time period covered, we compared a group of mothers with a history of prior psychiatric admission with those without a history of admission two or more years prior to the index admission. We compared those with and without a previous admission by diagnosis, Scottish Index of Multiple Deprivation (SIMD) quintile, age group, length of stay, time of admission and whether the mother had a previous pregnancy. SIMD score was used as a measure of social deprivation. The SIMD identifies small areas of multiple deprivation (data zones) across Scotland by combining 38 indicators across seven domains which are weighted. The domains include: current income (28%), employment (28%), health (14%), education (14%), geographic access to services (9%), crime (5%) and housing (2%) and are weighted based on evidence from Oxford University's Social Disadvantage Research Centre.[24] Individuals were divided into deprivation quintiles, depending on their deprivation score based on national averages.

We used logistic regression to describe the association between each variable and the risk of admission between those with a history of prior psychiatric admission and those without. ORs were adjusted for age and deprivation quintile or for age only (deprivation quintile) and deprivation quintile (age only).

Finally, to assess the impact of perinatal mental illness on early child development after childbirth, we generated a pragmatic indicator for 'at potential risk of adversity', defined as a child who was recorded as requiring intensive treatment at any time under the health plan indicators

(HPI) and/or who had no record of completing three doses of the 5-in-1 vaccine by 12 months. The HPI is assessed at first visit, at 6 to 8 weeks. For the HPI, three possible outcomes can be recorded by the health visitor for the level of care required ('core', 'additional' and 'intensive'). It should be noted that results should be interpreted cautiously as there has been inconsistency in the implementation of the HPI across health boards and others have not validated this measure. Please also note that the time of assessment is not related to the time of first admission and is not an assessment of the admission care.

## ETHICAL CONSIDERATIONS

Ethical approval for this study was obtained by the NHS Privacy Advisory Committee (PAC) (XRB12089).

## RESULTS

In total, 190 (11.0%) of this sample were admitted to a specialist MBU in Scotland. Table 1 highlights some important differences between MBU versus non-MBU admissions. First, women admitted to an MBU were significantly more likely to be admitted with non-affective psychosis (OR=1.97, 95% CI 1.22 to 3.18), affective psychosis (OR=2.44, 95% CI 1.37 to 4.33) and non-psychotic depressive episodes (OR=1.93, 95% CI 1.42 to 2.63). It is also notable that women admitted to an MBU were significantly more likely to live within affluent areas (and less likely to come from deprived areas) and in general more likely to be from an older age group (31–35, 36–40 and over 40) (table 1). Women admitted to an MBU were largely from the NHS Greater Glasgow and Clyde Health Board area (54.2% overall).

Table 1 also shows that MBU admissions were significantly less likely to have had a previous pregnancy and less likely to have had a previous psychiatric admission. It also highlights important differences in length of stay for MBU versus non-MBU admissions. It shows that women admitted to an MBU were significantly less likely to have brief stays in hospital (of 10 days or less) but significantly more likely to have stays of more than 10 days and more than twice as likely to have had a stay of 21–40 days or 40 days or more.

In total, 562 (32.5%) of the 1730 women had a history of psychiatric admission. Table 2 shows that this group were more likely to be admitted with a diagnosis of affective psychosis (7.3% vs 2.6%, adjusted OR=3.00, 95% CI 1.82 to 4.94) and non-affective psychosis (10.0% vs 6.4%, adjusted OR=1.55, 95% CI 1.07 to 2.25) but less likely to have an admission for a postpartum or non-psychotic depressive episode diagnosis. Those with a previous history of psychiatric admission were also significantly more likely to be located in the two most deprived quintiles and less likely to come from the two most affluent quintiles. Table 2 also shows they were less likely to be under 26 but significantly more likely to be from the

31–35, 36-40 and over 40 age groups. No significant differences were identified for length of stay, previous pregnancy or time of admission.

Table 3 shows that 518 (29.9%) of offspring were defined as being 'at potential risk of adversity' according to our criteria as assessed at the first health visitor visit after childbirth. Those at 'potential risk' were more likely (although not statistically significantly) to have had a mother with non-affective psychosis (OR=1.37, 95% CI 0.94 to 2.01) and significantly less likely to have a mother with a diagnosis of non-psychotic depression (OR=0.65, 95% CI 0.51 to 0.83). It is notable that they were significantly more likely to come from the most deprived quintiles and less likely to be from more affluent quintiles. No differences were found by age of mother nor by place of first admission either at a MBU or a non-MBU. Those children identified as 'at potential risk of adversity' were more likely to have had a mother with a history of a previous psychiatric admission.

## DISCUSSION

The aim of this study was to assess the extent to which sociodemographic factors and perinatal mental illness affects treatment and outcomes for both mothers and children. We covered two main areas: admissions to an MBU and factors associated with children identified as at 'potential risk of childhood adversity'.

In this large Scottish sample, women admitted to a one of the two Scottish MBUs (compared with women admitted to general psychiatry wards) were significantly more likely be diagnosed with a psychotic illness (non-affective psychotic illness or affective psychotic illness) and less likely to be admitted with other illnesses. There was no difference for early postpartum psychosis admissions between MBU and non MBU. This is in keeping with the notion that MBU admission is reserved for women suffering from the most serious mental disorders such as postpartum psychosis, mania, major depressive episodes with psychosis or schizophrenia.[18]

Women admitted to an MBU (compared with women admitted to general psychiatry wards) were more likely to live within affluent areas and in general more likely to be from an older age group (36–40 and over 40). It is possible that differences in sociodemographics of women accessing MBUs, might reflect a health inequality in terms of access to MBU admission. However this is a question, which requires further research.

To date, literature exploring accessibility to MBUs is limited. There is one recent systematic review investigating outcomes for women admitted to a mother and baby unit.[25] However accessibility to an MBU was not reported. There are other studies on women admitted to an MBU, but they give only results on the socioeconomic status of women and no objective marker of deprivation (such as SIMD) is usually included.[25] Studies investigating the link between deprivation and admission rate are limited. Some authors have described a link between

**Table 1** Comparison of characteristics of women admitted either with their infant to an MBU or without their infant to a psychiatric ward (non-MBU) (n=1729)

| | MBU admission | Non-MBU admission | OR, adjusted by age and deprivation (95% CI) |
|---|---|---|---|
| | Number (%) | Number (%) | |
| Total | 190 (11.0) | 1539 (89) | |
| Diagnosis | | | |
| Non-affective psychosis | 24 (12.6) | 107 (7.0) | 1.97 (1.22 to 3.18) |
| Affective psychosis | 18 (9.5) | 53 (3.4) | 2.44 (1.37 to 4.33) |
| Postpartum (0–12 weeks) psychosis | 18 (9.5) | 87 (5.7) | 1.56 (0.91 to 2.70) |
| Non-psychotic depressive episodes | 85 (44.7) | 447 (29.0) | 1.93 (1.42 to 2.63) |
| Any other diagnosis | 130 (68.4) | 1292 (84.0) | 0.45 (0.32 to 0.63) |
| Deprivation quintile | | | |
| Most deprived (1) | 64 (33.7) | 656 (43.4) | 0.68 (0.49 to 0.93) |
| 2 | 34 (17.9) | 336 (22.2) | 0.76 (0.51 to 1.13) |
| 3 | 28 (14.7) | 246 (16.3) | 0.88 (0.58 to 1.31) |
| 4 | 37 (19.5) | 169 (10.2) | 2.00 (1.35 to 2.96) |
| Least deprived (5) | 27 (14.2) | 105 (6.9) | 2.33 (1.49 to 3.65) |
| Age group | | | |
| Under 20 | 16 (8.4) | 172 (11.2) | 0.78 (0.45 to 1.34) |
| 20–25 | 40 (21.1) | 465 (30.2) | 0.65 (0.45 to 0.95) |
| 26–30 | 47 (24.7) | 407 (26.4) | 0.90 (0.63 to 1.29) |
| 31–35 | 50 (26.3) | 293 (19.0) | 1.38 (0.97 to 1.96) |
| 36–40 | 31 (16.3) | 170 (11.0) | 1.49 (1.03 to 1.58) |
| Over40 | 6 (3.2) | 32 (2.1) | 1.55 (0.63 to 3.80) |
| Length of admission | | | |
| 5 days or less | 35 (18.4) | 640 (41.6) | 0.34 (0.23 to 0.50) |
| 6–10 days | 25 (13.2) | 320 (20.8) | 0.59 (0.39 to 0.92) |
| 11–20 days | 42 (22.1) | 274 (17.8) | 1.27 (0.86 to 1.85) |
| 21–40 days | 49 (25.8) | 172 (11.2) | 2.51 (1.73 to 3.63) |
| 40 plus days | 39 (20.4) | 133 (8.6) | 2.48 (1.65 to 3.71) |
| Geographical location | | | |
| Greater Glasgow and Clyde | 103 (54.2) | 389 (25.3) | 3.50 (2.57 to 4.76) |
| Lothian | 55 (29.0) | 169 (11.0) | 3.30 (2.32 to 4.96) |
| Lanarkshire | 12 (6.3) | 169 (11.0) | 0.54 (0.29 to 1.00) |
| Any other health board | 20 (10.5) | 812 (52.7) | 0.02 (0.01 to 0.03) |
| Had previous pregnancy | 95 (50.0) | 935 (60.8) | 0.57 (0.41 to 0.78) |
| History of psychiatric admission | 43 (22.6) | 519 (33.7) | 0.57 (0.40 to 0.83) |
| Time of first admission | | | |
| Postpartum (0–12 weeks) | 102 (53.7) | 125 (8.1) | 11.96 (8.4 to 16.8) |

Women could have more than one diagnosis so numbers are greater than the number of women. In total 27 women had no Scottish Index of Multiple Deprivation quintile assigned due to missing or incorrect postcode and were excluded from the regression analysis for all variables. MBU, mother and baby unit.

psychiatric diagnoses, deprivation and admission rate[26] (in the general adult setting) while others have suggested a link between socioeconomic status and admission to forensic psychiatry services.[27] However, further detailed investigation in this area is required.

In our study, we found that women admitted to an MBU were more likely to have a longer average length of stay (40 plus days). This may be related to the need for a higher level of social functioning to ensure safe care for mothers and their babies on discharge from hospital.

**Table 2** Comparison of characteristics of women with and without a previous psychiatric admission prior to the index admission (n=1720)

| | Previous admission | No previous admission | OR adjusted by age and deprivation (95% CI) |
|---|---|---|---|
| | Number (% of total) | Number (% of total) | |
| Total | 562 (32.5) | 1168 (67.5) | |
| Diagnosis | | | |
| Non-affective psychosis | 56 (10.0) | 75 (6.4) | 1.55 (1.07 to 2.25) |
| Affective psychosis | 41 (7.3) | 30 (2.6) | 3.00 (1.82 to 4.94) |
| Postpartum psychosis | 57 (10.1) | 170 (14.6) | 0.36 (0.20 to 0.64) |
| Non-psychotic depressive episodes | 137 (24.4) | 395 (33.8) | 0.63 (0.50 to 0.79) |
| Any other diagnosis | 450 (80.1) | 973 (83.3) | 0.78 (0.60 to 1.02) |
| Deprivation quintile | | | |
| Most deprived | 251 (45.7) | 470 (40.7) | 1.31 (1.07 to 1.61) |
| 2 | 131 (23.9) | 239 (20.7) | 1.11 (0.87 to 1.42) |
| 3 | 85 (15.5) | 189 (16.4) | 0.90 (0.68 to 1.20) |
| 4 | 54 (9.8) | 152 (13.2) | 0.68 (0.49 to 0.95) |
| Least deprived | 28 (5.1) | 104 (9.0) | 0.48 (0.31 to 0.75) |
| Age group | | | |
| Under 20 | 37 (6.6) | 152 (13.0) | 0.44 (0.30 to 0.64) |
| 20–25 | 146 (26.0) | 359 (30.7) | 0.78 (0.62 to 0.98) |
| 26–30 | 160 (28.5) | 294 (25.2) | 1.19 (0.95 to 1.50) |
| 31–35 | 126 (22.4) | 217 (18.6) | 1.31 (1.01 to 1.68) |
| 36–40 | 80 (14.2) | 121 (10.4) | 1.45 (1.07 to 1.97) |
| Over 40 | 13 (2.3) | 25 (2.1) | 1.01 (0.50 to 2.03) |
| Length of stay | | | |
| 5 days or less | 203 (36.1) | 472 (40.4) | 0.87 (0.70 to 1.09) |
| 6–10 days | 112 (19.9) | 234 (20.0) | 0.99 (0.76 to 1.28) |
| 11–20 days | 104 (18.5) | 212 (18.2) | 0.95 (0.73 to 1.24) |
| 21–40 days | 71 (12.6) | 150 (12.8) | 0.99 (0.73 to 1.26) |
| 40 plus days | 72 (12.8) | 100 (8.6) | 1.50 (1.07 to 2.09) |
| Had previous pregnancy | 359 (63.9) | 671 (57.5) | 1.03 (0.82 to 1.29) |
| Time of first admission | | | |
| Postpartum (0–12 weeks | 57 (10.1) | 170 (14.6) | 0.71 (0.48 to 1.02) |

Women could have more than one diagnosis so numbers are greater than the number of women. In total, 27 women had no Scottish Index of Multiple Deprivation quintile assigned due to missing or incorrect postcode and were excluded from the regression analysis for all variables.

Further investigation comparing average length of stay in an MBU with particular focus on the cost-effectiveness of MBUs may be warranted.

Several notable findings arose from our comparison of mothers of children identified as 'at potential risk of adversity'. First, almost one-third (29%) of children fell into this category. This rate is similar to that reported by others. For example, Whitmore and colleagues[28] reported that social services were involved in 31.6% of all admissions to their MBU in Birmingham and 10% of admissions resulted in separate discharges.[28] Howard and colleagues[29] reported that 23% of women were discharged with their babies under some form of social services supervision.[29]

Second, our findings indicate that children in the 'at potential risk of adversity' group were more likely to come from deprived locations, have mothers with a previous psychiatric admission and have had a mother admitted with a non-affective psychosis (schizophrenia). This finding is also similar to that by others[18 28–30] who reported that women with a diagnosis of schizophrenia were discharged separately more often than other groups. Potential risk factors associated with risk of separation are complex but include: neonatal or infant medical problems or complications; maternal psychiatric disorder; paternal psychiatric disorder; maternal lack of good relationship with others; mother receipt of disability benefits and low social class.[30] In particular, schizophrenia,

**Table 3** Characteristics of mothers with children defined as being at 'potential risk of adversity' (n=1720)

| | At potential risk of adversity | Not at potential risk of adversity | OR adjusted by age and deprivation (95% CI) |
|---|---|---|---|
| | Number (% of total) | Number (% of total) | |
| Total | 518 (29.9) | 1212 (70.1) | |
| MBU stay | 58 (30.4) | 133 (69.6) | 1.18 (0.84 to 1.65) |
| General ward | 459 (29.8) | 1079 (70.2) | 1 |
| Diagnosis | | | |
| Non-affective psychosis | 49 (9.5) | 82 (6.7) | 1.37 (0.94 to 2.01) |
| Affective psychosis | 22 (4.3) | 49 (4.0) | 1.24 (0.73 to 2.12) |
| Postpartum psychosis | 21 (4.1) | 84 (6.9) | 0.62 (0.37 to 1.03) |
| Non-psychotic depressive episodes | 127 (24.5) | 405 (33.4) | 0.65 (0.51 to 0.83) |
| Any other diagnosis | 426 (82.2) | 997 (82.3) | 0.93 (0.70 to 1.22) |
| Deprivation quintile | | | |
| Most deprived | 274 (54.0) | 448 (37.5) | 1.88 (1.52 to 2.32) |
| 2 | 112 (22.1) | 227 (21.5) | 1.03 (0.80 to 1.33) |
| 3 | 61 (12.0) | 214 (17.9) | 0.61 (0.45 to 0.84) |
| 4 | 36 (7.1) | 169 (14.1) | 0.46 (0.32 to 0.68) |
| Least deprived | 24 (7.3) | 108 (9.3) | 0.51 (0.32 to 0.81) |
| Age group | | | |
| Under 20 | 61 (11.8) | 128 (10.6) | 1.02 (0.73 to 1.45) |
| 20–25 | 154 (29.7) | 351 (29.0) | 0.91 (0.72 to 1.15) |
| 26–30 | 154 (29.7) | 300 (24.8) | 1.32 (0.98 to 1.67) |
| 31–35 | 90 (17.4) | 253 (20.9) | 0.90 (0.69 to 1.17) |
| 36–40 | 47 (9.1) | 154 (12.7) | 0.73 (0.51 to 1.03) |
| Over40 | 12 (2.3) | 26 (2.2) | 1.15 (0.56 to 2.33) |
| Length of stay | | | |
| 5 days or less | 213 (41.1) | 456 (37.6) | 1.16 (0.93 to 1.44) |
| 6–10 days | 103 (19.9) | 244 (20.1) | 0.91 (0.70 to 1.19) |
| 11–20 days | 93 (18.0) | 223 (18.4) | 0.98 (0.73 to 1.29) |
| 21–40 days | 48 (11.2) | 166 (13.7) | 0.84 (0.60 to 1.17) |
| 40 plus days | 51 (9.9) | 123 (10.2) | 0.98 (0.68 to 1.40) |
| Had previous pregnancy | 319 (61.6) | 711 (58.9) | 1.11 (0.89 to 1.40) |
| History of psychiatric admission | 215 (41.5) | 347 (28.6) | 1.76 (1.41 to 2.21) |
| Time of first admission | | | |
| Postpartum (0–12 weeks | 73 (14.1) | 154 (12.7) | 1.31 (0.96 to 1.79) |

Women could have more than one diagnosis so numbers are greater than the number of women. In total 27 women had no Scottish Index of Multiple Deprivation quintile assigned due to missing or incorrect postcode and were excluded from the regression analysis for all variables. 'At potential risk of adversity' is defined as a child who was recorded as requiring intensive treatment at any time under the health plan indicators (HPI) and/or who had no record of completing three doses of the 5-in-1 vaccine by 12 months. HPI is assessed at first visit and at 6 to 8 weeks.
MBU, mother and baby unit.

personality disorder and poor social integration have all been related to poor clinical outcomes.[31]

Successful parenting is complex, but may be partly associated with family stability and access to financial and social resources.[32] While it is recognised that mothers with severe mental illness have specific needs relating to their children, their health and social care needs may not always be readily identified by healthcare professionals.[33]

In cases of separate discharge, additional support should be made available to minimise distress. Therefore taken together, our findings suggest that there may be some benefit to identifying mothers at particularly high risk in terms of adverse developmental outcomes for their children, such as mothers living in more deprived communities who have a history of psychiatric admissions for schizophrenia.

## STRENGTHS AND LIMITATIONS

Strengths of this study include the completeness of the sample, which was obtained from record linkage for the whole of Scotland. However, some limitations in this work are acknowledged. First, only psychiatric admission data were used, with no use of outpatient data. Our findings are therefore focused on the more severe end of the mental illness spectrum. Although we were able to determine if individuals had had a previous psychiatric admission, information about the number of previous admissions, family support or access to crisis teams were not available. The comparisons of women admitted to MBUs with women admitted to general psychiatric wards need to be interpreted cautiously. This is particularly in relation to the timing of admissions in relation to childbirth, as women will only be admitted to MBUs in the very late stages of pregnancy and in the first year postdelivery. The general psychiatric admissions included admissions up to 2 years predelivery and 2 years postdelivery and may therefore include women with long-term illnesses.

Furthermore, limited information about time and length stay at an MBU prevented any cost analysis being undertaken. Our definition of 'at potential risk of adversity' in offspring was a composite and pragmatic measure derived from the limited child health outcome data which was available to us from record linkage. There has also been some inconsistency in the implementation of the HPI across health boards. A more detailed and comprehensive assessment of child development in this group of mothers is therefore warranted.

## CONCLUSION

In conclusion, this study found that a health informatics/data linkage approach has considerable potential for improving our understanding of the social and clinical factors, which contribute to perinatal mental illness in mothers in Scotland, as well as potential adverse developmental outcomes for their children. To date there has been no systematic assessment of the benefits (or adverse effects) of specialised MBUs in Scotland. Given the current political–economic climate and the importance of early intervention, further research in this area would be of benefit.

**Contributors** GM undertook the data analyses. JLM and DJM wrote the initial draft of the paper. GM, RC and DJM were involved in the drafting of the paper.

**Funding** This project was funded by a CSO Health Informatics Call Grant.

**Competing interests** None declared.

**Ethics approval** NHS Privacy Advisory Committee (PAC) (XRB12089).

**Provenance and peer review** Not commissioned; externally peer reviewed.

**Data sharing statement** Data included in the study are available to the authors only.

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
