## [Reviewer comments · BMJ Open]

ARTICLE DETAILS

TITLE (PROVISIONAL)	Admission to Psychiatric Hospital for mental illnesses 2 years pre and post childbirth in Scotland: a health informatics approach to assessing mother and child outcomes.
AUTHORS	Langan, Julie; McLean, Gary; Martin, Daniel; Cantwell, Roch; Smith, Daniel

VERSION 1 - REVIEW

REVIEWER	Xiaoqin Liu The National Center for Register-based Research, Aarhus University, Denmark
REVIEW RETURNED	05-Apr-2017

GENERAL COMMENTS	The study by Martin et al. conducted a study based on the dataset from the Information Services Division (ISD) of NHS Scotland, which included all perinatal records between 2005–2009. This study comprised 3,290 pregnancy-related psychiatric admissions for 1,730 women. The authors aimed to investigate: 1) Risk factors for the admission to a specialist Mother and Baby Unit (MBU); 2) risk factors for readmission to a psychiatric hospital; and 3) the impact of perinatal mental illness on early childhood development. Linkage of Scottish datasets has contributed to our current knowledge in the field of perinatal psychiatry. For this reason, I welcome a study using these unique data sources. All these three research questions listed above are essential and would potentially add knowledge to this field. Although well-written, I found the study design difficult to understand. I read it several times and discussed with one of my colleagues. However, after this, I still was not sure how the study was designed. Overall, the method section lacked important information and needed details (see my comments below). I found all three sub-aims of importance, but could not help thinking that it might be too much to study three objectives within one manuscript. Moreover, it is also impossible to explore these three aims using the same settings. I would, therefore, suggest the authors to reconsider the objective of their study. Although it will be much work to do, it is very helpful to redesign this study from scratch. Additionally, I have some specific questions, comments and thoughts (mentioned in random order below) that the authors may wish to address. Major comments:
---

Comment 1: The investigation on MBU is particularly interesting, and there are few studies on this topic so far. However, as a non-Scottish researcher, I need more detailed explanations on how the system works, for instance, who ends up with MBU admission, who refers the women to the MBU, and do the women have a choice to be admitted to the MBU? In my opinion, a descriptive study on the characteristics of women with MBU admission is also important, in particular, its relation to childhood development. Whether children born to mothers with MBU admission have different health outcomes, compared to children of mothers with no MBU additions?

The authors explored the risk factors associated with admission to the MBU by comparing to women admitted to a general psychiatric ward. It is not explained why this reference group was selected. These two groups may not be entirely comparable such as family support and severity of the disorders, etc. (which is also partly mentioned by the authors). Also, as the authors discussed in the discussion section, MBU admissions may differ from admissions to a general psychiatric ward regarding the timing of admission in relation to childbirth. MBU admission is more likely to occur in late stages of pregnancy and the postpartum period, while childbirth is known to be a risk factor for severe psychiatric episodes. The interpretation of the finding is, therefore, difficult and needs to be cautious.

Comment 2: The authors mentioned “To explore possible factors associated with future readmission, we compared a group of mothers with a history of prior psychiatric admission to those without a history of admission two or more years prior to the index admission” (Page 5, Line 30–35). What was the definition of readmission? Please clarify. One of the most important risk factors for readmission is the number of previous admissions, which was not available in the dataset. The result on readmission is, therefore, very difficult to interpret.

Comment 3: The authors were unsure about their findings on childhood development. Has the indicator for “high risk of impaired development” been validated before? If the authors think that the definition of “high risk of developmental impairment” in offspring was not a valid measure, it might be better to drop this analysis or choose another definition.

Minor comments:

- 1) Abstract: Please add the statistical model in the abstract.
- 2) Page 5, Line 8: The authors mentioned that “This dataset has been described elsewhere”. I acknowledge this may be due to the word limit. However, a brief introduction of the dataset may be helpful.
- 3) There are some missing values in the variable deprivation quartile. Please clarify how the authors included these missing values in the models.
- 4) Page 5, Line 39–43: The authors used Cox regression model to calculate odds ratios. To my knowledge, hazard ratios, instead of odds ratios are calculated in the Cox models.
- 5) Page 5, Line 41–42: The authors mentioned that “Where possible, odds ratios were adjusted for age and deprivation quintile.” Were the other covariates mutually adjusted for in all the models? Please clarify.
- 6) Page 6, Line 10–11: “and non-psychotic depressive episodes (OR=1.93, 95% CI: 1.42–1.63)”. This may be a typo. Please correct it accordingly.

REVIEWER	Nine GLANGEAUD INSERM, Paris, France
REVIEW RETURNED	11-Apr-2017

GENERAL COMMENTS	This study is an analysis of retrospective data from the Scottish maternity record (SMR02) linked to the Scottish Psychiatric admission record (SMR04) to identify factors associated to admission either to mother-baby units (MBU) or to psychiatric hospital and also the factors related to psychiatric readmission and to some negative outcomes for children at 6 to 8 weeks after birth. The study period was 2003-2011, defined as two years before and two years after the index childbirth, which takes place within the years 2005 to 2009. 1730 women were included, corresponding to 3290 pregnancies during the study period (2005-2009). General comments The strong side of this study is the quite big, national sample with linkage between two databases, allowing description of psychiatric admission related to childbirth. The weak side of the study is the information available in the two databases which doesn't allow detailed descriptions and limit the type of factors that could be studied. For instance, women's psychiatric diagnostics are not precise and are given as broad groups of diagnoses. Although the category "any other diagnostic" represents about 80% of the sample there is no detail on diagnostics included. Some diagnoses may be over represented in the description of the sample, as in more than 100 cases there was several diagnosis reported for the same woman. Is there information available on parity in the data base or only information on "previous pregnancy" (yes or no)? During the entire study period of 4 years, some women had several pregnancies and several psychiatric admissions. What information is available when readmission occurs? Is it pre or post childbirth? Only 125 women, over 1539 women, were admitted at non-MBU admission during the postpartum period (0-12 weeks). When the others 1406 women were admitted? Same question is for MBU admission of 88 women not in this postpartum period. Women having chronic or long term pathologies may be over represented specially in the non-MBU sample. Comparison of MBU and non MBU admission At a MBU, women are admitted with their infant, usually during the first year after childbirth and sometimes already during the late pregnancy. If I understand well the results women admitted alone at a psychiatric department (Non-MBU) may be admitted any time during the study period of two years after childbirth (not to be called "postpartum period", even in research). Non-MBU admission would be better compared to MBU admission for the same post-childbirth period. Moreover, MBU care includes not only psychiatric mother's care, but also child's care and support for the mother-child relationship and
--

safety development of the child. This type of care in MBUs need time to be done. Those differences of care should to be described already in the introduction and also discussed.

Child outcome, what as the authors call "child high-risk of developmental impairment" is "a composite and pragmatic measure derived from the limited child health outcome data". Those data were collected, at the first home visit, by health visitors reporting about "child requiring intensive care or/ and no record of completing three doses of the 5 in 1 vaccine by 12 months". Please change 'high risk of developmental impairment' to "intensive level of child care required" or something more close to what you have assessed. What other information is available on children outcome in the database?

Because of those limits of data available, authors need to be more cautious and more self-critic when describing their results and discussing them.

The introduction

The references cited in the introduction are not enough updated

Your introduction should focus on the background of your main results on:

- Care and context of admission in MBU for women with their infant and in psychiatric hospital without the child in Scotland (See some suggestions of international references on MBUs at the end of the review)
- Prevalence of relapse and maternal psychiatric readmission before and after childbirth according to maternal pathology
- Outcomes for the child according to maternal pathology and psychosocial factors.

Please correct the sentence "Since then the UK has acted as a leader in this field" by adding "and also Australia and France" (see references at the end of the review)."

For non-Scottish readers, please give more detail on The Scottish Index of Multiple Deprivation (SIMD) and also on geographical area (affluent quintiles). Are those regions different in terms of deprivation and mental health resources?

Methods

Some comments about method are described above about the need to compare MBU and non-MBU within the same period after childbirth.

The other limitation for testing risk factors is the relatively small number of women admitted to MBU during the study period which doesn't allow testing many different factors with many classes. I would suggest that age groups, deprivation quintile and length of stay should be tested with 2 or 3 sib-group, to be statistically more powerful.

It is not clear for me how is define the "index admission" for non-MBU admissions? Is it the childbirth, or any time before and/or after childbirth?

Results general comment

Is there differences in results, according to the year considered for index admission, if yes, can it be discuss in relation to changes in the mental or perinatal health policies in Scotland, during the study period?

Are odds ratios different after adjustment on age and/or on deprivation than before adjustment?

Please describe in the results (not in the discussion) more precisely diagnosis distribution within the broad diagnostic categories: “non-affective psychosis”, “affective psychosis” and finally “any other diagnosis” which represents 68 to 84% of the sample. You have about 127 women with multiple diagnoses; this may lead to over represent some diagnosis. This should be discussed.

You speak of “at least one readmission”, please describe when readmissions take place: Is it only during the one or two years after the index childbirth and does it include “history of prior admission”. Is it readmission in the same structure (MBU or non-MBU) than the index admission or not?

Please give information about the differences of characteristics of women (age, diagnosis, and deprivation), according admission (or readmission) during 0-12 weeks postpartum compared to other times.

What is the difference of diagnosis of the women staying longer at MBU than those at non MBU hospitals? MBU long stay may be also due also to worries for the child safety and not only for women's mental health. This may be discussed.

In table 4 please give the percentage of women's admission at a MBU or at a psychiatric hospital, according to child outcome (intensive level of child care required or not). This may give interesting information to discuss.

For the results on geographical area (affluent quintiles), please add when necessary (results not shown in tables)

Tables

The titles of tables are not in agreement with their content and please remove table 3.

Explain in a note to tables what your definition of “deprivation quintile” and write fully (SIMD): The Scottish Index of Multiple Deprivation.

Table 1

The title may be: Comparison of characteristics of women admitted either with their infant at a Mother-Baby unit (MBU) or without their infant in a psychiatric department (Non-MBU) (N=1729)

Last column title of table 1, I guess that the odds ratios for deprivation quintile or for age are not adjusted on the respective variable.

Table 2

The title may be: Comparison of characteristics of women either or not with previous admission to the index admission (N=1720)..

Table 3

Content of Table 3 on length of stay and readmission may be described in the text. To my point of view there is no meaning to compare readmission according to length of admission, as the care and aims in MBU and non MBU are quite different and timing may differ due to the infant needs in MBU admission.

Table 4

Title may be : Comparison of characteristics of women “level of child care required” as reported by health visitor at the first postpartum (home?) visit (N=1720)

Put a note about your definition of “intensive level of child care required”: assessed by a Health Visitor at both 10-day and 6-8 week child-health checks as who report on a child as requiring intensive treatment at any time under the Health Plan Indicators (HPI) and/or who had no record of completing three doses of the 5 in 1 vaccine by 12 months was generated.

Discussion (see some suggestions of studies that may be discussed at the end of the review)

Several comments have been already given above.

Several epidemiologic studies have shown that past psychiatric history is one of the main risk-factor for postpartum psychiatric admission after child birth. It is good that you have confirmed this point on a big national sample of women who gave birth in Scotland. Several studies have also shown the importance of psychosocial factors. Your results on socioeconomic deprivation and geographic location are interesting too and may be more discussed.

Discussion on the relation between socioeconomic difficulties and type of pathology is interesting. It was discussed for schizophrenic mother by Abel et al. but it is difficult to know if it is a cause or a consequence of the pathology.

Note for the discussion that past history of psychiatric admission, chronicity and readmission may be associated with specific diagnosis and they may interact with deprive context and psychosocial factors regardless of the location of admission MBU or not.

Note also that women, who are addressed to MBU, need to be able to be cared also for their relationship with the child and child needs for safety should be considered before having MBU admission. Those criteria, for addressing a woman to a MBU, may lead to different profiles of women than those of women admitted to general hospital, without their child.

Strength and limitations should be more discussed (see all the comments above).

Conclusion should be written in a more careful way, considering all the comments above.

My suggestions:

Even when there is a need for separation, for child protection or for other reasons, support has to be given to the mother and the child, to try to reduce separation trauma, as done in MBUs.

You may also comment on Women's satisfaction describe in the literature for MBU care.

There is a need for more studies about cost differences between MBU and non-MBU, according to short term and long term outcome benefice for the mother and for the child.

Some suggestions of studies for the introduction or discussion

Milgrom J (2015) Impact of parental psychiatric illness on infant development. In: Sutter-Dallay A-L, Glangeaud-Freudenthal NM-C, Guedeney A, Riecher-Rössler A (eds) Joint care of parents and infants in perinatal psychiatry. Springer,47-78.

Munk-Olsen T, Laursen TM, Pedersen CB et al (2006) New parents and mental disorders: a population-based registered study. JAMA 296:2582–2589

Description of MBU admission and care

Ian Brockington, Ruth Butterworth, Nine Glangeaud-Freudenthal. An international position paper on mother-infant (perinatal) mental health, with guidelines for clinical practice Arch Womens Ment Health (2017) 20:113–120

Glangeaud-Freudenthal Nine M.-C., Louise Howard & Anne-Laure Sutter-Dallay. Treatment – Mother-Infant inpatient units In: Perinatal Mental Illness: Guidance for the Obstetrician-Gynecologist Ed Michael O'Hara, Katherine Wisner and Jerry Joseph, USA Best Practice & Research Clinical Obstetrics and Gynaecology 28 (2014) 147–157), <http://dx.doi.org/10.1016/j.bpobgyn.2013.08.015>

Sutter-Dallay et al. Eds Joint Perinatal Psychiatric Care for Parents and Infants. Springer, 2015. ISBN 9783319215570 •9783319215563. DOI 10.1007/978-3-319-21557-0 <http://www.springer.com/us/book/9783319215563>

MBU maternal and infant outcome from MBUs

Glangeaud-Freudenthal et al (2013) Predictors of infant foster care in cases of maternal psychiatric disorders. Social Psychiatry and Psychiatric Epidemiology 48(4); 553-561

Abel KM, Webb RT, Salmon MP, Wan MW, Appleby L (2005) Prevalence and predictors of parenting outcomes in a cohort of mothers with schizophrenia admitted for joint mother and baby psychiatric care in England. J Clin Psychiatry 66:781–789.

Howard L, Shah N, Salmon M, Appleby L (2003) Predictors of social services supervision of babies of mothers with mental illness after admission to a psychiatric mother and baby unit. Soc Psychiatry Psychiatr Epidemiol 38:450–455 DOI 10.1007/s00737-016-0684-7

For the discussion

	Howard L, Flach C, Leese M et al (2010) Effectiveness and cost-effectiveness of admissions to women's crisis houses compared with traditional psychiatric wards: pilot patient-preference randomised controlled trial. Br J Psychiatry 197:32–40. Howard LM, Hunt K (2008) The needs of mothers with severe mental illness: a comparison of assessments of needs by staff and patients. Arch Womens Ment Health 11:131–136. Neil S, Sanderson H, Wieck A (2006) A satisfaction survey of women admitted to a psychiatric mother and baby unit in the northwest of England. Arch Womens Ment Health 9:109–112. Sutter-Dallay AL, Murray L, Dequae-Merchadou L et al (2011) A prospective longitudinal study of the impact of early postnatal vs. chronic maternal depressive symptoms on child development. Eur Psychiatry 26:484–489
--	--

VERSION 1 – AUTHOR RESPONSE

Reviewer: 1

1.1 The study by Martin et al. conducted a study based on the dataset from the Information Services Division (ISD) of NHS Scotland, which included all perinatal records between 2005–2009. This study comprised 3,290 pregnancy-related psychiatric admissions for 1,730 women. The authors aimed to investigate: 1) Risk factors for the admission to a specialist Mother and Baby Unit (MBU); 2) risk factors for readmission to a psychiatric hospital; and 3) the impact of perinatal mental illness on early childhood development.

Linkage of Scottish datasets has contributed to our current knowledge in the field of perinatal psychiatry. For this reason, I welcome a study using these unique data sources. All these three research questions listed above are essential and would potentially add knowledge to this field.

Although well-written, I found the study design difficult to understand. I read it several times and discussed with one of my colleagues. However, after this, I still was not sure how the study was designed. Overall, the method section lacked important information and needed details (see my comments below).

RESPONSE:

We acknowledge that the methods section could have been clearer and have amended this. Please see Pages 5 and 6.

“We used a dataset from the Information Services Division (ISD) of NHS Scotland, which included all maternity records (SMR04) between 2005-2009, linked to all psychiatric hospital admission records (SMR02) between 2003-2011. This dataset has been described elsewhere (Langan Martin et al., 2016). However, in brief, for each maternity record any psychiatric admission was reported by week for the 104 weeks pre-childbirth and post-childbirth. Admission types were defined by ICD-10 codes:

psychosis-only admissions included 'non-affective psychosis' (F20, F20.3, F20.5, F20.6, F20.8, F20.9, F21X, F22.0, F22.8, F22.9, F23.0, F23.1, F23.2, F23.3, F23.8, F23.9, F24X, F28X, F29X), 'affective psychosis' (F25.0, F25.1, F25.2, F25.9, F30.2, F31.2, F31.5, F32.3, F33.3) and 'postpartum psychosis' (F53.0, F53.1, F53.9); admissions due to a non-psychotic depressive episode included F32.0, F32.00, F32.01, F32.1, F32.10, F32.11, F32.2, F32.8, F32.9, F33.0, F33.00, F33.1, F33.10, F33.11, F33.2, F33.4, F33.8, and F33.9. For the category of 'other admissions' we included all other recorded psychiatric ICD-10 codes".

I found all three sub-aims of importance, but could not help thinking that it might be too much to study three objectives within one manuscript. Moreover, it is also impossible to explore these three aims using the same settings. I would, therefore, suggest the authors to reconsider the objective of their study. Although it will be much work to do, it is very helpful to redesign this study from scratch. Additionally, I have some specific questions, comments and thoughts (mentioned in random order below) that the authors may wish to address.

Major comments:

1.2 The investigation on MBU is particularly interesting, and there are few studies on this topic so far. However, as a non-Scottish researcher, I need more detailed explanations on how the system works, for instance, who ends up with MBU admission, who refers the women to the MBU, and do the women have a choice to be admitted to the MBU? In my opinion, a descriptive study on the characteristics of women with MBU admission is also important, in particular, its relation to childhood development. Whether children born to mothers with MBU admission have different health outcomes, compared to children of mothers with no MBU admissions?

RESPONSE:

We agree with the reviewer's comments that the pathway to MBU admission could be clearer in our manuscript. We have therefore added in the below to provide a more detailed explanation as to the pathway to MBU admission. Please see Page 4

"In the West of Scotland, women in the post-natal period who are the primary carer of their baby and thought to require psychiatric admission to hospital are discussed and where possible assessed by the local Consultant Perinatal Psychiatrist. Hospital admission is arranged if required and in some cases, compulsory treatment under the Mental Health Act is necessary."

1.3 The authors explored the risk factors associated with admission to the MBU by comparing to women admitted to a general psychiatric ward. It is not explained why this reference group was selected. These two groups may not be entirely comparable such as family support and severity of the disorders, etc. (which is also partly mentioned by the authors). Also, as the authors discussed in the discussion section, MBU admissions may differ from admissions to a general psychiatric ward regarding the timing of admission in relation to childbirth. MBU admission is more likely to occur in late stages of pregnancy and the postpartum period, while childbirth is known to be a risk factor for severe psychiatric episodes. The interpretation of the finding is, therefore, difficult and needs to be cautious.

The authors mentioned "To explore possible factors associated with future readmission, we compared a group of mothers with a history of prior psychiatric admission to those without a history of admission two or more years prior to the index admission" (Page 5, Line 30–35). What was the definition of readmission? Please clarify. One of the most important risk factors for readmission is the number of previous admissions, which was not available in the dataset. The result on readmission is, therefore, very difficult to interpret.

RESPONSE:

Thank you for the above comment. This variable has now been removed from the table and text as it confusing to the reader.

1.4 The authors were unsure about their findings on childhood development. Has the indicator for “high risk of impaired development” been validated before? If the authors think that the definition of “high risk of developmental impairment” in offspring was not a valid measure, it might be better to drop this analysis or choose another definition.

RESPONSE:

While we acknowledge that this indicator has not been validated elsewhere, we felt it was a pragmatic definition given the information available. We have therefore altered the terminology used, to reflect this indicator: ‘at potential risk of adversity’. Please see Page 2 onwards.

We have also added in a statement reinforcing the idea that results should be cautiously interpreted. Please see Page 5

“It should be noted that results should be interpreted cautiously as there has been inconsistency in the implementation of the HPI across health boards and others have not validated this measure.”

Minor comments:

1.5 Abstract: Please add the statistical model in the abstract.

RESPONSE:

This has now been included in the Abstract. Please see Page 2.

“Logistic regression was used to describe the association between each variable and the risk of admission between those with a history of prior psychiatric admission and those without.”

1.6 Page 5, Line 8: The authors mentioned that “This dataset has been described elsewhere”. I acknowledge this may be due to the word limit. However, a brief introduction of the dataset may be helpful.

RESPONSE:

We agree with the reviewer that a brief description of the dataset would be helpful. We have therefore included more detail on the dataset in the Methods Section. Please see Page 5.

“This dataset has been described elsewhere (Langan Martin et al., 2016). However, in brief, for each maternity record any psychiatric admission was reported by week for the 104 weeks pre-childbirth and post-childbirth. Admission types were defined by ICD-10 codes: psychosis-only admissions included ‘non-affective psychosis’ (F20, F20.3, F20.5, F20.6, F20.8, F20.9, F21X, F22.0, F22.8, F22.9, F23.0, F23.1, F23.2, F23.3, F23.8, F23.9, F24X, F28X, F29X), ‘affective psychosis’ (F25.0, F25.1, F25.2,

F25.9, F30.2, F31.2, F31.5, F32.3, F33.3) and 'postpartum psychosis' (F53.0, F53.1, F53.9); admissions due to a non-psychotic depressive episode included F32.0, F32.00, F32.01, F32.1, F32.10, F32.11, F32.2, F32.8, F32.9, F33.0, F33.00, F33.1, F33.10, F33.11, F33.2, F33.4, F33.8, and F33.9. For the category of 'other admissions' we included all other recorded psychiatric ICD-10 codes".

1.7 There are some missing values in the variable deprivation quartile. Please clarify how the authors included these missing values in the models.

RESPONSE:

Individuals with missing deprivation scores were excluded from all the regression analysis. The tables have now been amended to reflect this. Please see Tables in Pages 11 -14.

1.8 Page 5, Line 39–43: The authors used Cox regression model to calculate odds ratios. To my knowledge, hazard ratios, instead of odds ratios are calculated in the Cox models.

RESPONSE:

This was an error and has now been corrected. Please see Page 6

"We used logistic regression to describe the association between each variable and the risk of admission between those with a history of prior psychiatric admission and those without".

1.9 Page 5, Line 41–42: The authors mentioned that "Where possible, odds ratios were adjusted for age and deprivation quintile." Were the other covariates mutually adjusted for in all the models? Please clarify.

RESPONSE:

This has been clarified. Please see Page 6.

"Odds ratios were adjusted for age and deprivation quintile or for age only (deprivation quintile) and deprivation quintile (age only)"

1.10 Page 6, Line 10–11: "and non-psychotic depressive episodes (OR=1.93, 95% CI: 1.42–1.63)". This may be a typo. Please correct it accordingly.

RESPONSE:

This was an error and has now been changed. Please see Page 6.

" non-psychotic depressive episodes (OR=1.93, 95% CI: 1.42–2.63)".

REVIEWER 2:

This study is an analysis of retrospective data from the Scottish maternity record (SMR02) linked to the Scottish Psychiatric admission record (SMR04) to identify factors associated to admission either to mother-baby units (MBU) or to psychiatric hospital and also the factors related to psychiatric readmission and to some negative outcomes for children at 6 to 8 weeks after birth The study period

was 2003-2011, defined as two years before and two years after the index childbirth, which takes place within the years 2005 to 2009.
1730 women were included, corresponding to 3290 pregnancies during the study period (2005-2009).

General comments

The strong side of this study is the quite big, national sample with linkage between two databases, allowing description of psychiatric admission related to childbirth.

The weak side of the study is the information available in the two databases which doesn't allow detailed descriptions and limit the type of factors that could be studied.

2.1 For instance, women's psychiatric diagnostics are not precise and are given as broad groups of diagnoses. Although the category "any other diagnostic" represents about 80% of the sample there is no detail on diagnostics included.

Some diagnoses may be over represented in the description of the sample, as in more than 100 cases there was several diagnosis reported for the same woman.

RESPONSE:

We agree with the reviewer that a brief description of the "any other diagnostic category" would be helpful. We have therefore included this information in the Methods Section. Please see Page 5.

"For the category of 'other admissions' we included all other ICD-10 codes recorded."

2.2 Is there information available on parity in the data base or only information on "previous pregnancy" (yes or no)?

RESPONSE:

Information on parity was not available. The only information was if women had a previous pregnancy.

2.3 During the entire study period of 4 years, some women had several pregnancies and several psychiatric admissions. What information is available when readmission occurs? Is it pre or post childbirth?

RESPONSE:

It is correct that during the 4 year period, some women may have had several pregnancies. Information is available of what time the admissions occurs (two years pre or post birth). The variable "readmission" been removed from the table and text as it confusing to the reader.

2.4 Only 125 women, over 1539 women, were admitted at non-MBU admission during the postpartum period (0-12 weeks). When the others 1406 women were admitted? Same question is for MBU admission of 88 women not in this postpartum period. Women having chronic or long term pathologies may be over represented specially in the non-MBU sample.

RESPONSE:

The majority of women were admitted either before pregnancy or between 13 and 104 weeks after delivery in the non-MBU sample. It is possible that long-term pathologies may be included in the non-MBU sample. This has therefore been acknowledged as a limitation in the strengths and limitations section. Please see Page 9.

"The general psychiatric admissions included admissions up to 2 years pre partum and 2 years post-

partum and may therefore include women with long-term illnesses”

2.5 Comparison of MBU and non MBU admission

At a MBU, women are admitted with their infant, usually during the first year after childbirth and sometimes already during the late pregnancy. If I understand well the results women admitted alone at a psychiatric department (Non-MBU) may be admitted any time during the study period of two years after childbirth (not to be called “postpartum period”, even in research).

Non-MBU admission would be better compared to MBU admission for the same post-childbirth period.

RESPONSE:

The above is noted and the terms changed to Non-MBU Admission and MBU Admission to aid clarity. The term post-partum period has been removed from the keywords.

2.6 Moreover, MBU care includes not only psychiatric mother’s care, but also child’s care and support for the mother-child relationship and safety development of the child. This type of care in MBUs need time to be done. Those differences of care should to be described already in the introduction and also discussed.

RESPONSE

We agree with the reviewer that more detail regarding the type of care delivered in MBUs should be described. We have therefore amended the manuscript to reflect this. Please see Page 4

“Currently in the UK where possible, postpartum women (and those in later pregnancy) with severe mental illnesses such as psychosis or severe depressive disorder are admitted to a specialised Mother and Baby Unit (MBU). In the West of Scotland, women in the post-natal period who are the primary carer of their baby and thought to require psychiatric admission to hospital are discussed and where possible assessed by the local Consultant Perinatal Psychiatrist. The current Scottish Intercollegiate Guidelines Network (SIGN) clinical guidelines for the management of perinatal mood disorder reflect this (SIGN, 2012).

MBUs are highly specialised, expensive and limited resources, where expertise in both treatment of psychiatric disorders and child care are required (Glangeaud-Freudenthal et al., 2014).”

2.7 Child outcome, what as the authors call “child high-risk of developmental impairment“ is “a composite and pragmatic measure derived from the limited child health outcome data”. Those data were collected, at the first home visit, by health visitors reporting about “child requiring intensive care or/ and no record of completing three doses of the 5 in 1 vaccine by 12 months”. Please change ‘high risk of developmental impairment’ to “intensive level of child care required” or something more close to what you have assessed. What other information is available on children outcome in the database?

RESPONSE:

We have altered the terminology used to better reflect the pragmatic indicator used. Please see Page 2 onwards ‘at potential risk of adversity’.

No additional information on child outcome was included in the database.

2.8 Because of those limits of data available, authors need to be more cautious and more self-critic when describing their results and discussing them.

RESPONSE:

We agree with the reviewer’s comments that the dataset has limitations and so the results need to be cautiously interpreted. We have therefore included a statement reinforcing this. Please see Page 5

“It should be noted that results should be interpreted cautiously as there has been inconsistency in the implementation of the HPI across health boards and others have not validated this measure.”

The introduction

2.9 The references cited in the introduction are not enough updated. Your introduction should focus on the background of your main results on:

RESPONSE:

This introduction has been amended to include more up to date references. Please see Pages 4 and 5.

“The joint admission of mentally ill mothers and their infants was pioneered by Thomas Main in 1948 (Brockington, 1996). Since then the UK, Australia and France have acted as leaders in this field (Brockington et al., 2017; Cazas and Glangeaud-Freudenthal, 2004). Currently in the UK where possible, postpartum women (and those in later pregnancy) with severe mental illnesses such as psychosis or severe depressive disorder are admitted to a specialised Mother and Baby Unit (MBU). In the West of Scotland, women in the post-natal period who are the primary carer of their baby and thought to require psychiatric admission to hospital are discussed and where possible assessed by the local Consultant Perinatal Psychiatrist. Hospital admission is arranged if required and in some cases, compulsory treatment under the Mental Health Act is necessary. The current Scottish Intercollegiate Guidelines Network (SIGN) clinical guidelines for the management of perinatal mood disorder reflect this (SIGN, 2012). In our previous, study (Langan Martin et al., 2016) we found that compared with the pre-pregnancy period, admission rates fell during pregnancy, increased markedly during the early postpartum period (0 to 6 weeks), and remained elevated for 2 years after childbirth. Within the most affluent quintile, admission Incidence rate ratios (IRRs) were higher in the early postpartum period (IRR=1.29, 95% CI 1.02 to 1.59) than in the late postpartum period (IRR=0.87, 95% CI 0.74 to 0.98).

MBUs are highly specialised, expensive and limited resources, where expertise in both treatment of psychiatric disorders and child care are required (Glangeaud-Freudenthal et al., 2014). Although there are currently 15 in England and 2 in Scotland (Royal College of Psychiatrists Quality Network for Perinatal Mental Health Services, 2016), access to this specialised service is poorer in many other High Income Countries (HICs; such as the US and Canada) and Low And Middle Income Countries (LAMICs). Given the importance of adverse childhood experiences (ACEs) in future health (Bellis et al., 2015, Felitti et al., 1998), perinatal mental illness and the potential impact it has on offspring is a priority area for research and practice.”

2.10 Please correct the sentence “Since then the UK has acted as a leader in this field” by adding “and also Australia and France” (see references at the end of the review).”

RESPONSE:

This has been altered. Please see Page 4.

“The joint admission of mentally ill mothers and their infants was pioneered by Thomas Main in 1948 (Brockington, 1996). Since then the UK, Australia and France have acted as leaders in this field (Brockington et al., 2017; Cazas and Glangeaud-Freudenthal, 2004).”

2.11 For non-Scottish readers, please give more detail on The Scottish Index of Multiple Deprivation (SIMD) and also on geographical area (affluent quintiles). Are those regions different in terms of deprivation and mental health resources?

RESPONSE:

We agree with the reviewer that for Non-Scottish Readers more information on SIMD would be helpful. This has now been included. Please see Page 6

“SIMD score was used as a measure of social deprivation. The SIMD identifies small areas of multiple deprivation (datazones) across Scotland by combining 38 indicators across 7 domains which are weighted. The domains include: current income (28%), employment (28%), health (14%), education (14%), geographic access to services (9%), crime (5%), and housing (2%) and are weighted based on evidence from Oxford University’s Social Disadvantage Research Centre (SIMD, Scotland).”

The average datazone identified covers on average 500 people. Affluent areas can therefore be located next to deprived areas and both can in urban or rural places. Regarding deprivation status and mental health resources, it is beyond the scope of this paper to assess differences in mental health provision however this would be an area of potential further research.

Methods

2.12 Some comments about method are described above about the need to compare MBU and non-MBU within the same period after childbirth. The other limitation for testing risk factors is the relatively small number of women admitted to MBU during the study period which doesn’t allow testing many different factors with many classes. I would suggest that age groups, deprivation quintile and length of stay should be tested with 2 or 3 sib-group, to be statistically more powerful.

RESPONSE:

Thank you for the above comment. While we agree with the reviewer that sub-group analyses would have been interesting, it was felt that this would reduce the statistical power of the model.

2.13 It is not clear for me how is define the “index admission” for non-MBU admissions? Is it the childbirth, or any time before and/or after childbirth?

RESPONSE:

Thank you for the above comment the index admission is defined as the first admission in the period from two years before and after childbirth.

Results general comment

2.14 Is there differences in results, according to the year considered for index admission, if yes, can it be discuss in relation to changes in the mental or perinatal health policies in Scotland, during the study period?

RESPONSE:

Thank you for the above comment. While we agree with the reviewer that sub-group analyses by index year of admission would have been interesting, the numbers were too small for this to be viable.

2.15 Are odds ratios different after adjustment on age and/or on deprivation than before adjustment?

RESPONSE:

Thank you for the above comment. There was little difference in odds ratios before or after adjustment.

2.16 Please describe in the results (not in the discussion) more precisely diagnosis distribution within the broad diagnostic categories: “non-affective psychosis”, “affective psychosis” and finally “any other diagnosis” which represents 68 to 84% of the sample. You have about 127 women with multiple diagnoses; this may lead to over represent some diagnosis. This should be discussed.

RESPONSE:

Thank you for the above comment. The phenomenon of diagnostic instability has been noted and included in the results section and the limitations section. Please see Page 7 and Page 10.

“It is notable that there was some evidence of diagnostic instability in this cohort for women admitted to both an MBU and non-MBU.” And

“There was also evidence of diagnostic instability within the dataset.”

2.17 You speak of “at least one readmission”, please describe when readmissions take place: Is it only during the one or two years after the index childbirth and does it include “history of prior admission”. Is it readmission in the same structure (MBU or non-MBU) than the index admission or not?

RESPONSE:

Thank you for the above comment. This variable has now been removed from the table and text as it confusing to the reader.

2.18 Please give information about the differences of characteristics of women (age, diagnosis, and deprivation), according admission (or readmission) during 0-12 weeks postpartum compared to other times.

RESPONSE:

Thank you for the above comment. The baseline characteristics for the above women were described in our previous paper, published in BMJ Open. Please see:
Langan-Martin, J., McLean, G., Cantwell, R., and Smith, D. J. (2016) Admission to psychiatric hospital in the early and late post-partum periods: Scottish national linkage study. *BMJ Open*, 6(1), e008758. (doi:10.1136/bmjopen-2015-008758) (MID:26733566)

2.19 What is the difference of diagnosis of the women staying longer at MBU than those at non MBU hospitals? MBU long stay may be also due also to worries for the child safety and not only for women’s mental health. This may be discussed.

RESPONSE:

The discussion has been amended based on the comments above.

2.20 In table 4 please give the percentage of women's admission at a MBU or at a psychiatric hospital, according to child outcome (intensive level of child care required or not). This may give interesting information to discuss.

RESPONSE:

We thank the reviewer for this suggestion and have now amended the table to include this information. Please see Table 3 Page 13.

At potential risk of adversity Not at potential risk of adversity Odds ratio adjusted by age and deprivation (95%CI)

Number (% of total) Number (% of total)

Total 518 (29.9) 1,212 (70.1)

MBU Stay 58 (30.4) 133 (69.6) 1.18 (0.84-1.65)

General Ward 459 (29.8) 1,079 (70.2)

2.21 For the results on geographical area (affluent quintiles), please add when necessary (results not shown in tables)

RESPONSE:

Thank you for the above. A SIMD score was not calculated for a whole Health board such as Greater Glasgow and Clyde, Lothian etc. More detail on SIMD has been included in the paper. Please see explanation of SIMD included in the Methods Section Page 6

"SIMD score was used as a measure of social deprivation. The SIMD identifies small areas of multiple deprivation (datazones) across Scotland by combining 38 indicators across 7 domains which are weighted. The domains include: current income (28%), employment (28%), health (14%), education (14%), geographic access to services (9%), crime (5%), and housing (2%) and are weighted based on evidence from Oxford University's Social Disadvantage Research Centre (SIMD, Scotland)."

Tables

2.22 The titles of tables are not in agreement with their content and please remove table 3.

RESPONSE:

We agree with the reviewer that Table 3 is not necessary. It has been removed. The titles of the remaining tables altered in line with the suggestions below.

2.23 Explain in a note to tables what your definition of "deprivation quintile" and write fully (SIMD): The Scottish Index of Multiple Deprivation.

RESPONSE:

Regarding "deprivation quintile" individuals, were divided into quintiles related to their SIMD score. Please see Methods Section on Page 6.

"Individuals were divided into deprivation quintiles, depending on their deprivation score."

Scottish Index of Multiple Deprivation (SIMD) has been written out in full in all the Tables. Please see Pages 11-13

Table 1

2.24 The title may be: Comparison of characteristics of women admitted either with their infant at a Mother-Baby unit (MBU) or without their infant in a psychiatric department (Non-MBU) (N=1729)

RESPONSE:

The title of table 1 has been amended. Please see page 11.

“Table 1: Comparison of characteristics of women admitted either with their infant to a Mother-Baby Unit (MBU) or without their infant to a psychiatric ward (Non-MBU) (N=1729)”

2.25 Last column title of table 1, I guess that the odds ratios for deprivation quintile or for age are not adjusted on the respective variable.

RESPONSE:

Thank you for this comment. Clarification of this has been included in the Methods Section. Please see Page 6.

“Odds ratios were adjusted for age and deprivation quintile or for age only (deprivation quintile) and deprivation quintile (age only).”

Table 2

2.26 The title may be: Comparison of characteristics of women either or not with previous admission to the index admission (N=1720).

RESPONSE:

The title of Table 2 has been amended. Please see Page 12.

“Table 2. Comparison of characteristics of women with and without a previous psychiatric admission prior to the index admission (N=1720).”

Table 3

2.27 Content of Table 3 on length of stay and readmission may be described in the text. To my point of view there is no meaning to compare readmission according to length of admission, as the care and aims in MBU and non MBU are quite different and timing may differ due to the infant needs in MBU admission.

RESPONSE:

We agree with the reviewer that Table 3 is not necessary. It has been removed. Information pertaining length of stay has been included in Table 1.

Table 4

2.28 Title may be : Comparison of characteristics of women “level of child care required” as reported by health visitor at the first postpartum (home?) visit (N=1720)

RESPONSE:

The title of Table 4 (now Table 3) has been amended. Please see Page 13.

“Table 3. Characteristics of mothers with children defined as being at ‘potential risk of adversity’ (N=1720)”

2.29 Put a note about your definition of “intensive level of child care required”: assessed by a Health Visitor at both 10-day and 6-8 week child-health checks as who report on a child as requiring intensive treatment at any time under the Health Plan Indicators (HPI) and/or who had no record of completing three doses of the 5 in 1 vaccine by 12 months was generated.

RESPONSE:

A note regarding our definition of “at potential high risk of adversity” has been included. Please see Page 13.

Note: “at potential risk of adversity”, is defined as a child who was recorded as requiring intensive treatment at any time under the health plan indicators (HPI) and/or who had no record of completing three doses of the 5 in 1 vaccine by 12 months. HPI is assessed at first visit and at six to eight weeks.

2.30 Discussion (see some suggestions of studies that may be discussed at the end of the review)

RESPONSE:

The discussion has been amended based on the comments above. Please see Pages 7-9.

2.31 Strength and limitations should be more discussed (see all the comments above).

RESPONSE:

The strengths and limitations section has been amended based on the comments above. Please see Pages 9 and 10.

“Strengths of this study include the completeness of the sample, which was obtained from record linkage for the whole of Scotland. However, some limitations in this work are acknowledged. Firstly, only psychiatric admission data were used, with no use of out-patient data. Our findings are therefore focused on the more severe end of the mental illness spectrum. Although we were able to determine if individuals had had a previous psychiatric admission, information about the number of previous admissions, family support, or access to Crisis teams were not available. The comparisons of women admitted to MBUs with women admitted to general psychiatric wards needs to be interpreted cautiously. This is particularly in relation to the timing of admissions in relation to childbirth, as women will only be admitted to MBUs in the very late stages of pregnancy and in the first year post-partum. The general psychiatric admissions included admissions up to 2 years pre partum and 2 years post-partum and may therefore include women with long-term illnesses.

Furthermore, limited information about time and length stay at an MBU prevented any cost analysis being undertaken. Our definition of “at potential risk of adversity” in offspring was a composite and pragmatic measure derived from the limited child health outcome data which was available to us from record linkage. There has also been some inconsistency in the implementation of the HPI across health boards. A more detailed and comprehensive assessment of child development in this group of mothers is therefore warranted.”

2.32 Conclusion should be written in a more careful way, considering all the comments above.
RESPONSE:

The conclusion has been amended based on the comments above. Please see Page 10.

“In conclusion, this study found that a health informatics/data linkage approach has considerable potential for improving our understanding of the social and clinical factors, which contribute to perinatal mental illness in mothers in Scotland, as well as potential adverse developmental outcomes for their children. To date there has been no systematic assessment of the benefits (or adverse effects) of specialised MBUs in Scotland. Given the current political-economic climate and the importance of early intervention, further research in this area would be of benefit”.

2.33 My suggestions:

Even when there is a need for separation, for child protection or for other reasons, support has to be given to the mother and the child, to try to reduce separation trauma, as done in MBUs.

RESPONSE:

We agree with the reviewer that this is an important area of consideration. This has now been included in the discussion. Please see Page 9

“In cases of separate discharge, additional support should be made available to minimise distress”

2.34 You may also comment on Women’s satisfaction describe in the literature for MBU care.

RESPONSE:

This has now been included. Please see Page 4.

“Although there are currently 15 in England and 2 in Scotland (Royal College of Psychiatrists Quality Network for Perinatal Mental Health Services, 2016), access to this specialised service is poorer in many other High Income Countries (HICs; such as the US and Canada) and Low And Middle Income Countries (LAMICs) despite women appearing to be satisfied with this type of care (Neil et al., 2005)”.

2.35 There is a need for more studies about cost differences between MBU and non-MBU, according to short term and long term outcome benefice for the mother and for the child.

RESPONSE:

We agree with the reviewer that this is an important area for future work. This has now been included in the conclusion. Please see Page 10.

“Given the current political-economic climate and the importance of early intervention, further research in this area would be of benefit.”

2.36 Some suggestions of studies for the introduction or discussion

RESPONSE:

Thank you for these suggested references. The majority have been included. Further details are included below.

2.37 Milgrom J (2015) Impact of parental psychiatric illness on infant development. In: Sutter-Dallay A-L, Glangeaud-Freudenthal NM-C, Guedeney A, Riecher-Rössler A (eds) Joint care of parents and infants in perinatal psychiatry. Springer,47-78.

RESPONSE:

This study has now been included. Please see Page 4.

2.38 Munk-Olsen T, Laursen TM, Pedersen CB et al (2006) New parents and mental disorders: a population-based registered study. *JAMA* 296:2582–2589

RESPONSE:

This study has now been included. Please see Page 4.

2.39 Description of MBU admission and care

Ian Brockington, Ruth Butterworth, Nine Glangeaud-Freudenthal. An international position paper on mother-infant (perinatal) mental health, with guidelines for clinical practice *Arch Womens Ment Health* (2017) 20:113–120

RESPONSE:

This study has now been included. Please see Page 4.

2.40 Glangeaud-Freudenthal Nine M.-C., Louise Howard & Anne-Laure Sutter-Dallay. Treatment – Mother-Infant inpatient units In: *Perinatal Mental Illness: Guidance for the Obstetrician-Gynecologist* Ed Michael O’Hara, Katherine Wisner and Jerry Joseph, USA Best Practice & Research Clinical Obstetrics and Gynaecology 28 (2014) 147–157, <http://dx.doi.org/10.1016/j.bpobgyn.2013.08.015>

RESPONSE:

This study has now been included. Please see Page 4.

2.41 Sutter-Dallay et al. Eds *Joint Perinatal Psychiatric Care for Parents and Infants*. Springer, 2015. ISBN 9783319215570 •9783319215563. DOI 10.1007/978-3-319-21557-0 <http://www.springer.com/us/book/9783319215563>

RESPONSE:

This study has now been included. Please see Page 4.

2.42 MBU maternal and infant outcome from MBUs

Glangeaud-Freudenthal et al (2013) Predictors of infant foster care in cases of maternal psychiatric disorders. *Social Psychiatry and Psychiatric Epidemiology* 48(4); 553-561

RESPONSE:

This study has now been included. Please see Page 9.

2.43 Abel KM, Webb RT, Salmon MP, Wan MW, Appleby L (2005) Prevalence and predictors of parenting outcomes in a cohort of mothers with schizophrenia admitted for joint mother and baby psychiatric care in England. *J Clin Psychiatry* 66:781–789.

RESPONSE:

This study has now been included. Please see Page 9.

2.44 Howard L, Shah N, Salmon M, Appleby L (2003) Predictors of social services supervision of babies of mothers with mental illness after admission to a psychiatric mother and baby unit. *Soc*

RESPONSE:

This study has now been included. Please see Page 9.

2.45 For the discussion

Howard LM, Hunt K (2008) The needs of mothers with severe mental illness: a comparison of assessments of needs by staff and patients. Arch Womens Ment Health 11:131–136.

RESPONSE:

This study has now been included. Please see Page 9.

2.46 Neil S, Sanderson H, Wieck A (2006) A satisfaction survey of women admitted to a psychiatric mother and baby unit in the northwest of England. Arch Womens Ment Health 9:109–112.

RESPONSE:

This study has now been included. Please see Page 4.

2.47 Sutter-Dallay AL, Murray L, Dequae-Merchadou L et al (2011) A prospective longitudinal study of the impact of early postnatal vs. chronic maternal depressive symptoms on child development. Eur Psychiatry 26:484–489

RESPONSE:

This study has now been included. Please see Page 4.

We feel this paper has now been substantially improved.

VERSION 2 – REVIEW

REVIEWER	Nine M-C Glangeaud INSERM, Paris
REVIEW RETURNED	16-Jun-2017

GENERAL COMMENTS	The manuscript is much improved. However there are still minor corrections requested. - Some typing mistakes in the document and in the references. For those mistakes please below with the notes.- Some mistakes or incomplete information in describing and discussing results (describe below) Introduction “To identify factors associated with: admission to a specialist Mother and Baby Unit (MBU), and the impact of perinatal mental illness on early childhood development..” you don't really measure an impact suggestion: " Mother and Baby Unit (MBU), and risk for early childhood development in the context of a pregnancy-related psychiatric admissions..."
--

“They were more likely to come from affluent areas (OR: 2.33 95%CI 1.49-3.65). “

The fact that they more often live close to the UMB is not of great interest I think.

On the contrary your result on deprive area is interesting and original to my opinion. if you have to choose, I would suggest to give only this result (see also my comments below). Therefore, I would suggest to replace or to add the following sentence:

“They are less likely to be from the most deprived areas. (OR 0.68 95%CI 0.49-0.93)”

Method section

on early child development assessment
please add “... eight weeks after childbirth”.

please add: The time of assessment is not related to the time of first admission or it is not an assessment of the admission care.

Results

Table 1

Add the results on age not in the discussion but here..... “It shows that women admitted to an MBU were significantly more likely to be from an older age 36-40 and less from younger age 20-25 and less likely to have brief stays in hospital.....”

Table 2

Correct under 26 and not 25 years

Table 3

Add “Table 3 shows that 518 (29.9%) of offspring were defined as being “at potential risk of adversity, according to our criteria assessed at the first Health Visitor visit after childbirth.”

Add “No differences were found by age of mother nor by place of first admission either at a MBU or a non-MBU.

Those results has been added to the table 3 are interesting original results that should be in the results.

Discussion

Add for readers not so aware of MUB setting

“In this large Scottish sample, women admitted to one or the two Scottish MBUs with their infant (compared to women admitted alone to general psychiatry wards).....”

You compare the pathologies distribution in MBU and non MBU

Don't give results in the discussion don't give results in the discussion and also there is a mistake OR: 1397??

Please add “In this large Scottish sample, women admitted to two Scottish MBU (compared to women admitted to general psychiatry wards) were significantly more likely be diagnosed with a psychotic illness (non-affective psychotic illness or affective psychotic illness

and less of other type of pathologies. No difference for early post-partum psychosis admissions between MBU and non MBU." it is important to mention also negative results.

"This is in keeping with the notion that MBU admission is reserved for women suffering from the most serious mental disorders" Also in psychiatric wards you have women with severe or serious mental disorders. What do you mean by "the most serious?"

"Women admitted to an MBU were also more likely to live within affluent areas (and less likely to come from deprived areas) and in general more likely to be from an older age group (31-35, 36-40 and over 40). It is possible that this might reflect a health inequality in terms of access to MBU admission but this is a question, which requires further research."

Why you say "and less likely to come from deprived areas "under (..)? it is a very important result.

Please correct" ... in general more likely to be from an older age..." 31-35 and over 40 are not significant from your results please remove those two.

My suggestion would be to only mention "from an older age. ".(see previous comment in results to add the details there).

What "this" refer to in the sentence: "...this might reflect a health inequality in terms of access to MBU admission but this is a question, which requires further research."

Do you mean : the deprived areas have less referrals to MBUs or that MBUs are less located in deprived area?

Please clarify for non-Scottish readers who are not aware of the context of MBUs location.

ref 25 is not the only review on MBUs outcomes

"There is a recent systematic review investigating outcomes for women admitted to a mother and baby unit [25]

My suggestion to complete the discussion with results that comfort your hypothesis fby adding: and several studies on broad MBUs sample showing that, for women with schizophrenia, being from a high social class may be protector (Abel et al 2005) and risk factors independently associated with mother-infant separation were not only related to infant problems and parental psychiatric disorder but also if mother get disability benefits and belong to a low social class (Glangeaud et al 2013) and poor social integration (as measured by occupational status) was related to poor clinical outcomes for women's mental health (Glangeaud et al 2011) see abstracts below.

Add: "Accessibility to an MBU was not reported, and main studies on MBU give only results on the socio-economic status of women no objective marker of deprivation (such as SIMD) was included[25]. »

Add that the results you are reporting here are not for MBUs "Some authors have described, in studies done at general psychiatric hospitals, a link between psychiatric diagnoses, deprivation and admission rate"

How you define severe presentation? "This may be a consequence of a more severe presentation requiring more intensive support..." What results support your hypothesis?

At MBUs, long stay is when it is needed for the mother-child

relationships and for safety of the child to find the best solution at discharge. If the maternal pathology is really very severe and acute she will be referred to a psychiatric ward and will not stay on the MBU, this also happen when the mother can't benefit from the presence of her child and don't seem to become able to invest the relationship with her child even with the support of the staff.

“Several notable findings arose from our comparison of mothers of children identified as “at potential risk of adversity...”
Please add comments also on publications from MBUs in France and Belgium relevant to your discussion.

Glangeaud et al 2013 Predictors of infant foster care in cases of maternal psychiatric disorders

Soc Psychiatry Psychiatr Epidemiol (2013) 48:553–561

DOI 10.1007/s00127-012-0527-

Herewith the Abstract

Purpose Our aim was to investigate the factors associated with mother–child separation at discharge, after joint hospitalization in psychiatric mother–baby units (MBUs) in France and Belgium.

Because parents with postpartum psychiatric disorders are at risk of disturbed parent–infant interactions, their infants have an increased risk of an unstable early foundation. They may be particularly vulnerable to environmental stress and have a higher risk of developing some psychiatric disorders in adulthood.

Methods This prospective longitudinal study of 1,018 women with postpartum psychiatric disorders, jointly admitted with their infant to 16 French and Belgian psychiatric mother–baby units (MBUs), used multifactorial logistic regression models to assess the risk factors for mother–child separation at discharge from MBUs. Those factors include some infant characteristics associated with personal vulnerability, parents’ pathology and psychosocial context.

Results: Most children were discharged with their mothers, but 151 (15 %) were separated from their mothers at discharge. Risk factors independently associated with separation were: (1) neonatal or infant medical problems or complications; (2) maternal psychiatric disorder; (3) paternal psychiatric disorder; (4) maternal lack of good relationship with others; (5) mother receipt of disability benefits; (6) low social class.

Conclusions: This study highlights the existence of factors other than maternal pathology that lead to decisions to separate mother and child for the child’s protection in a population of mentally ill mothers jointly hospitalized with the baby in the postpartum period.

Also the abstract from

Glangeaud et al European Psychiatry 26 (2011) 215–223

doi:10.1016/j.eurpsy.2010.03.006

Purpose: This study assessed the underexplored factors associated with significant improvement in mothers’ mental health during postpartum inpatient psychiatric care.

Methods: This study analyzed clinical improvement in a prospective cohort of 869 women jointly admitted with their infant to 13 psychiatric Mother-Baby Units (MBUs) in France between 2001 and 2007. Predictive variables tested were: maternal mental illness (ICD-10), sociodemographic characteristics, mental illness and childhood abuse history, acute or chronic disorder, pregnancy and birth data, characteristics and mental health of the mother’s partner, and MBU characteristics.

Results: Two thirds of the women improved significantly by discharge. Admission for 25% was for a first acute episode very

	early after childbirth. Independent factors associated with marked improvement at discharge were bipolar or depressive disorder, a first acute episode or relapse of such an episode. Schizophrenia, a personality disorder, and poor social integration (as measured by occupational status) were all related to poor clinical outcomes. Discussion: Most women improved significantly while under care in MBUs. Our results emphasize the importance of the type of disease but also its chronicity and the social integration when providing postpartum psychiatric care. If you need, I can send you reprints. I hope that my comments have helped you improving your very interesting analysis. Looking forward to reading more publications from you.
--	---

VERSION 2 – AUTHOR RESPONSE

The manuscript is much improved. However there are still minor corrections requested.

2.1 Introduction

“To identify factors associated with: admission to a specialist Mother and Baby Unit (MBU), and the impact of perinatal mental illness on early childhood development..”

you don't really measure an impact suggestion:

" Mother and Baby Unit (MBU), and risk for early childhood development in the context of a pregnancy-related psychiatric admissions..."

Response:

This has been altered as suggested above. Please see Page 5:

“The aim of the study was to use a data linkage approach to investigate factors associated with admission to a specialist Mother and Baby Unit (MBU) and the risk to early childhood development in the context of a pregnancy related psychiatric admission.”

2.2 “They were more likely to come from affluent areas (OR: 2.33 95%CI 1.49-3.65). “

The fact that they more often live close to the UMB is not of great interest I think.

On the contrary your result on deprive area is interesting and original to my opinion. if you have to choose, I would suggest to give only this result (see also my comments below). Therefore, I would suggest to replace or to add the following sentence:

“They are less likely to be from the most deprived areas. (OR 0.68 95%CI 0.49-0.93)”

Response:

This has been altered as suggested above. Please see Page 3:

“They were less likely to come from deprived areas (OR: 0.68 95%CI 0.49-0.93)”.

2.3 Method section

on early child development assessment

please add “:... eight weeks after childbirth”.

Response:

This has been altered. To state “after childbirth” Please see Page 6:

“Finally, to assess the impact of perinatal mental illness on early child development after childbirth, we generated a pragmatic indicator for “at potential risk of adversity”, defined as a child who was recorded as requiring intensive treatment at any time under the health plan indicators (HPI) and/or who had no record of completing three doses of the 5 in 1 vaccine by 12 months.”

2.4 please add: The time of assessment is not related to the time of first admission or it is not an assessment of the admission care.

Response:

This has been altered as suggested above. Please see Page 6:

“Please also note that the time of assessment is not related to the time of first admission and is not an assessment of the admission care.”

2.5 Results

Table 1

Add the results on age not in the discussion but here..... “It shows that women admitted to an MBU were significantly more likely to be from an older age 36-40 and less from younger age 20-25 and less likely to have brief stays in hospital.....”

Response:

This has been altered as suggested above. Please see Page 7:

“Table 2 also shows they were less likely to be under 26 but significantly more likely to be from the 31-35,36-40 and over 40 age groups.”

2.6 Table 2

Correct under 26 and not 25 years

Response:

This has been altered as suggested above. Please see Page 7:

“Table 2 also shows they were less likely to be under 26 but significantly more likely to be from the 31-35 and 36-40 age groups.”

2.7 Table 3

Add “Table 3 shows that 518 (29.9%) of offspring were defined as being “at potential risk of adversity, according to our criteria assessed at the first Health Visitor visit after childbirth.”

Response:

This has been altered as suggested above. Please see Page 7:

“Table 3 shows that 518 (29.9%) of offspring were defined as being “at potential risk of adversity” according to our criteria as assessed at the first Health Visitor visit after childbirth”.

2.8 Add “No differences were found by age of mother nor by place of first admission either at a MBU or a non-MBU.

Those results has been added to the table 3 are interesting original results that should be in the results.

Response:

This has been altered as suggested above. Please see Page 7:

“No differences were found by age of mother nor by place of first admission either at a MBU or a non-MBU”

2.9 Discussion

Add for readers not so aware of MUB setting

"In this large Scottish sample, women admitted to one or the two Scottish MBUs with their infant (compared to women admitted alone to general psychiatry wards)....."

Response:

This has been further clarified in the Methods Section on Page 5:

"There are two MBUs in Scotland, the West of Scotland Mother and baby Unit (Leverndale Hospital, NHS Greater Glasgow and Clyde) and Mental Health Mother and Baby Unit, St John's Hospital, (West Lothian) where mothers are admitted with their baby in the post-partum period. For women admitted to a general psychiatric hospital this is generally without their baby."

2.10 You compare the pathologies distribution in MBU and non MBU

Don't give results in the discussion don't give results in the discussion and also there is a mistake OR: 1397??

Response:

Thank you for noting this. This has been removed from the Discussion Section.

2.11 Please add "In this large Scottish sample, women admitted to two Scottish MBU (compared to women admitted to general psychiatry wards) were significantly more likely be diagnosed with a psychotic illness (non-affective psychotic illness or affective psychotic illness and less of other type of pathologies. No difference for early post-partum psychosis admissions between MBU and non MBU." it is important to mention also negative results.

Response:

This has been further detailed as suggested above. Please see Page 8:

"In this large Scottish sample, women admitted to a one of the two Scottish MBUs (compared to women admitted to general psychiatry wards) were significantly more likely be diagnosed with a psychotic illness (non-affective psychotic illness or affective psychotic illness) and less likely to be admitted with other illnesses. There was no difference for early post-partum psychosis admissions between MBU and non MBU. This is in keeping with the notion that MBU admission is reserved for women suffering from the most serious mental disorders such as postpartum psychosis, mania, major depressive episodes with psychosis or schizophrenia [18]."

2.12 "This is in keeping with the notion that MBU admission is reserved for women suffering from the most serious mental disorders"

Also in psychiatric wards you have women with severe or serious mental disorders. What do you mean by "the most serious?"

Response:

This has been further clarified on Page 8:

"This is in keeping with the notion that MBU admission is reserved for women suffering from the most serious mental disorders such as postpartum psychosis, mania, major depressive episodes with psychosis or schizophrenia [18]."

2.13 "Women admitted to an MBU were also more likely to live within affluent areas (and less likely to come from deprived areas) and in general more likely to be from an older age group (31-35, 36-40

and over 40). It is possible that this might reflect a health inequality in terms of access to MBU admission but this is a question, which requires further research.”
Why you say “and less likely to come from deprived areas “under (..)? it is a very important result.

Response:

This has been further clarified on Page 8:

“Women admitted to an MBU (compared to women admitted to general psychiatry wards) were more likely to live within affluent areas and in general more likely to be from an older age group (36-40 and over 40).”

2.14 Please correct” ... in general more likely to be from an older age...” 31-35 and over 40 are not significant from your results please remove those two.

My suggestion would be to only mention “from an older age. ”.(see previous comment in results to add the details there).

Response:

This has been changed, please see Page 8:

“Women admitted to an MBU (compared to women admitted to general psychiatry wards) were..... in general more likely to be from an older age group (36-40 and over 40).”

2.15 What “this” refer to in the sentence: “...this might reflect a health inequality in terms of access to MBU admission but this is a question, which requires further research.”

Do you mean : the deprived areas have less referrals to MBUs or that MBUs are less located in deprived area?

Response:

This has been further clarified, please see Page 8:

“It is possible that differences in socio-demographics of women accessing MBUs, might reflect a health inequality in terms of access to MBU admission. However this is a question, which requires further research.”

2.16 Please clarify for non-Scottish readers who are not aware of the context of MBUs location.

Response:

This has been further clarified in the Methods Section on Page 5:

“There are two MBUs in Scotland, the West of Scotland Mother and baby Unit (Leverndale Hospital, NHS Greater Glasgow and Clyde) and Mental Health Mother and Baby Unit, St John’s Hospital, (West Lothian) where mothers are admitted with their baby in the post-partum period. For women admitted to a general psychiatric hospital this is generally without their baby.”

2.17 ref 25 is not the only review on MBUs outcomes

“There is a recent systematic review investigating outcomes for women admitted to a mother and baby unit [25]

My suggestion to complete the discussion with results that comfort your hypothesis by adding: and several studies on broad MBUs sample showing that, for women with schizophrenia, being from a high social class may be protector (Abel et al 2005) and risk factors independently associated with mother-infant separation were not only related to infant problems and parental psychiatric disorder but also if mother get disability benefits and belong to a low social class (Glangeaud et al 2013) and poor social integration (as measured by occupational status) was related to poor clinical outcomes for women’s mental health (Glangeaud et al 2011) see abstracts below.

Response:

This has been modified. Please see Page 9:

“Secondly, our findings indicate that children in the “at potential risk of adversity” group were more likely to come from deprived locations, have mothers with a previous psychiatric admission and have had a mother admitted with a non-affective psychosis (schizophrenia). This finding is also similar to that by others [18,28-30] who reported that women with a diagnosis of schizophrenia were discharged separately more often than other groups. Potential risk factors associated with risk of separation are complex, but include: neonatal or infant medical problems or complications; maternal psychiatric disorder; paternal psychiatric disorder; maternal lack of good relationship with others; mother receipt of disability benefits; and low social class[30]. In particular schizophrenia, personality disorder, and poor social integration have all been related to poor clinical outcomes[31]”.

2.18 Add: “Accessibility to an MBU was not reported, and main studies on MBU give only results on the socio-economic status of women no objective marker of deprivation (such as SIMD) was included[25]. »

Response:

This has been further clarified Please see Page 8:

“To date literature exploring accessibility to MBUs is limited. There is one recent systematic review investigating outcomes for women admitted to a mother and baby unit[25]. However accessibility to an MBU was not reported. There are other studies on women admitted to an MBU, but they give only results on the socio-economic status of women and no objective marker of deprivation (such as SIMD) is usually included[25].”

2.19 Add that the results you are reporting here are not for MBUs

“Some authors have described, in studies done at general psychiatric hospitals, a link between psychiatric diagnoses, deprivation and admission rate”

Response:

This important point has been made more explicit. Please see Page 8:

“Some authors have described a link between psychiatric diagnoses, deprivation and admission rate[26] (in the general adult setting)”

2.20 How you define severe presentation? “This may be a consequence of a more severe presentation requiring more intensive support...” What results support your hypothesis?

At MBUs, long stay is when it is needed for the mother-child relationships and for safety of the child to find the best solution at discharge. If the maternal pathology is really very severe and acute she will be referred to a psychiatric ward and will not stay on the MBU, this also happen when the mother can't benefit from the presence of her child and don't seem to become able to invest the relationship with her child even with the support of the staff.

Response:

We acknowledge this important viewpoint and have altered the text as below. Please see Page 9:

“This may be related to the need for a higher level of social functioning to ensure safe care for mothers and their babies on discharge from hospital.”

2.21 “Several notable findings arose from our comparison of mothers of children identified as “at potential risk of adversity...”

Please add comments also on publications from MBUs in France and Belgium relevant to your

discussion.

Response:

We acknowledge these important references and these have now been included into the discussion.

Please see Page 9:

“Secondly, our findings indicate that children in the “at potential risk of adversity” group were more likely to come from deprived locations, have mothers with a previous psychiatric admission and have had a mother admitted with a non-affective psychosis (schizophrenia). This finding is also similar to that by others [18,28-30]who reported that women with a diagnosis of schizophrenia were discharged separately more often than other groups. Potential risk factors associated with risk of separation are complex, but include: neonatal or infant medical problems or complications; maternal psychiatric disorder; paternal psychiatric disorder; maternal lack of good relationship with others; mother receipt of disability benefits; and low social class[30]. In particular schizophrenia, personality disorder, and poor social integration have all been related to poor clinical outcomes[31]”.